# *Where's Whaledo*: A software toolkit for array localization of animal vocalizations

**Eric R. Snyder**\*, **Alba Solsona-Berga, Simone Baumann-Pickering, Kait E. Frasier, Sean M. Wiggins, John A. Hildebrand**

Scripps Institution of Oceanography, University of California San Diego, La Jolla, California, United States of America

\* snyder.eric.r@gmail.com

**Data Availability Statement:** All code is publicly available at github.com/ericSnyderSIO/ wheresWhaledo All data used in the paper are

## Abstract

*Where's Whaledo* is a software toolkit that uses a combination of automated processes and user interfaces to greatly accelerate the process of reconstructing animal tracks from arrays of passive acoustic recording devices. Passive acoustic localization is a non-invasive yet powerful way to contribute to species conservation. By tracking animals through their acoustic signals, important information on diving patterns, movement behavior, habitat use, and feeding dynamics can be obtained. This method is useful for helping to understand habitat use, observe behavioral responses to noise, and develop potential mitigation strategies. Animal tracking using passive acoustic localization requires an acoustic array to detect signals of interest, associate detections on various receivers, and estimate the most likely source location by using the time difference of arrival (TDOA) of sounds on multiple receivers. *Where's Whaledo* combines data from two small-aperture volumetric arrays and a variable number of individual receivers. In a case study conducted in the Tanner Basin off Southern California, we demonstrate the effectiveness of *Where's Whaledo* in localizing groups of *Ziphius cavirostris*. We reconstruct the tracks of six individual animals vocalizing concurrently and identify *Ziphius cavirostris* tracks despite being obscured by a large pod of vocalizing dolphins.

## Author summary

Reconstructing the movement of animals from their vocalizations is a powerful method to observe their behavior in situations where visual monitoring is impractical. Arrays of acoustic recording devices can be used to determine the location of vocalizing animals and a series of locations can be linked to form tracks. However, reconstructing tracks requires methods of determining which animal in a group is vocalizing, finding the same vocalization on multiple recording devices, and determining the most likely location of the animal based on the relative times the sound arrived at various recording devices. We have developed a toolkit called *Where's Whaledo* to assist researchers in reconstructing the behavior of these animals using arrays of acoustic recording devices. This toolkit greatly accelerates the process of reconstructing their tracks using a combination of automated processes and user interfaces. We use *Where's Whaledo* to reconstruct the tracks of

being uploaded to https://doi.org/10.5061/dryad.c866t1gfj.

**Funding:** Funding provided by the Office of Naval Research, Grant#: N00014-15-2587 to SBP and by the Pacific Fleet, Grant#: N00014-19-1-2583, Grant 12849596 to JAH. The funders had no role in study design, data collection and analysis, decision to publish, or preparation of the manuscript.

**Competing interests:** The authors have declared that no competing interests exist.

deep-diving beaked whales (*Ziphius cavirostris*). We successfully reconstruct tracks of groups of up to five whales vocalizing concurrently.

## Introduction

Passive acoustic monitoring (PAM) has been increasingly utilized to monitor animals in the wild [1–3]. The use of arrays of acoustic sensors has further enabled the localization of animal sounds, providing additional avenues of research including the study of behavior and a better understanding of animal population dynamics [4, 5]. Acoustic sensing has advantages over other common methods that are dependent on observers having suitable weather and lighting conditions to carry out visual surveys. PAM provides a method for non-invasive, long-term observations.

Cetaceans in particular are difficult to directly observe, but they produce species-specific vocalizations for both navigation and communication [2, 6]. Arrays of acoustic recording devices can be deployed to collect continuous data for months, providing a non-invasive method for studying cetacean behavior and presence. This method has become essential for studying deep-diving cetacean species, like beaked whales (family *Ziphiidae*), Sperm whales (*Physeter macrocephalus*), Risso's Dolphins (*Grampus griseus*), and pilot whales (genus *Globicephala*), which are pelagic and often spend relatively little time at the surface [7–11]. PAM has provided valuable insights into their behavior despite their elusiveness [7, 12–16].

For deep-divers, PAM is emerging as an essential method for studying their population structure and dynamics [17–19]. This requires *a priori* knowledge of a number of features, like group size, vocalization rates, and acoustic detection ranges and probabilities. While some studies have estimated these parameters using acoustic models or information known about closely related species or populations, obtaining direct measurements for a specific species and site would likely improve the estimates [17]. Most of these features can be estimated by reconstructing tracks from acoustic data. Group sizes can be estimated by identifying the number of individual tracks in an encounter. Detection ranges and probabilities can be estimated based on the positions of detected animals. Additionally, passive acoustic localization can provide valuable information about depths and durations of dives, foraging depths and behaviors, responses to anthropogenic sounds or other environmental stressors, and insights into potential harm mitigation strategies.

Passive acoustic localization of cetacean vocalizations using arrays of hydrophones has been used to reconstruct tracks of a number of cetacean species, like beaked whales, common dolphins (*Delphinus delphis*), and sperm whales (*Physeter macrocephalus*) ([13–16, 20–25]. Different approaches to localization have been implemented for different configurations of hydrophones, and to observe different species or behaviors of interest. Many of these studies have used localization to reconstruct two-dimensional approximations of tracks, either horizontal tracks [26, 27] or depth and range to the instrument [14, 23]. Three-dimensional localizations have been obtained using an individual hydrophone when accurate three-dimensional travel-time models could be constructed from measurements of sound speed profiles and bathymetry data [28].

Time difference of arrival (TDOA) localization uses the times a signal arrived at various receivers to estimate the location of a source. When receivers have sufficient coverage, a received signal can be localized in three-dimensional space. TDOA has been used to localize a number of vocalizing animals, including birds [29–31] bats [32], terrestrial animals [33], and aquatic animals [12, 13, 20, 34].

Reflections off the surface or refractions due to ray bending, called "multipath arrivals", can be used in localization [14, 28, 35–39]. Often, multpaths can be used to improve localizations or estimate the range between a source and an array [14, 38]. In cases where accurate models of multipath propagation can be made and there is significant azimuthal variation on these propagation patterns, the measured times of arrival of each multipath can be matched to models to estimate source locations in 3 dimensions from a single hydrophone [28, 37].

*Ziphius cavirostris* (*Zc*, colloquially referred to as goose-beaked whale or Cuvier's beaked whale) and common dolphins (*Delphinus delphis*) have been tracked in three dimensions using a small-aperture volumetric array [20]. The array contained four hydrophones in a tetrahedron configuration with $\approx 0.5$ m spacing between them. By measuring the TDOA between the hydrophones, the Direction Of Arrival (DOA) of the sound could be estimated as an azimuth and elevation angle to the animal. The most likely DOA was determined by minimizing the least squares error between model TDOAs and calculated TDOAs. By identifying differences in detection amplitude and azimuth angle, two individual *Zc* whales were tracked by assuming a constant dive speed.

Localization can be performed by combining both small-aperture and large-aperture TDOAs [13, 40, 41]. Gassmann *et al.* [13] demonstrated this embedded array approach by using two small-aperture volumetric arrays and three single-channel hydrophones to localize and track *Zc* offshore of Southern California. With these additional instruments, a total of 22 TDOAs could be used to estimate the location of a whale: six TDOAs each from two small-aperture arrays, and ten large-aperture TDOAs from five widely spaced instruments. This approach results in an overdetermined system which can improve estimation accuracy. However, uncertainty can be introduced due to ambiguous signal matching across widely spaced instruments. The difficulty increases as the number of sources increases, since the number of vocalizations arriving in the window of possible TDOAs also increases. To resolve this ambiguity, Gassmann *et al.* [13] plotted all possible TDOAs and manually identified the most likely correct TDOA from these sequences. They then used a maximum likelihood equation to determine the model location that best fit the measured TDOAs, successfully localizing a total of 11 individual beaked whales in groups of up to three individuals vocalizing concurrently.

Methods of associating sources automatically are necessary for accelerating the localization process. One method for source association is to temporally align sequences of clicks on widely spaced receivers [12]. If the same pattern of clicks exists in multiple hydrophones, then these patterns can be aligned to determine which clicks arrived from each source.

Automated tracking methods are emerging which use advanced multi-target tracking algorithms to identify source associations, remove false detections, and estimate likely tracks using two volumetric arrays for encounters with simultaneous detections on both arrays. [16]. Due to the directional nature of many species' echolocation clicks [13, 42], simultaneous detections become increasingly uncommon as the distance between the instruments increases. Incorporating single-channel instruments, which are easier and cost less to deploy and recover, can increase the number of trackable encounters.

In this article, we provide a semi-automated method with opportunities for expert oversight to assist in the association of detections. We have developed a user-friendly MATLAB toolkit that builds on the methods of [20], [12], and [13] to assist researchers in obtaining tracks from acoustic datasets. To demonstrate the effectiveness of our toolkit, we used it to reconstruct $\approx$ 80 *Zc* tracks from a four-month deployment in the Southern California Bight. We were able to reconstruct tracks for groups of up to five individuals vocalizing concurrently, a significant improvement over previous methods. We also addressed several challenges in preparing datasets for localization, including determining instrument locations and array orientations, synchronizing clocks, and calculating uncertainties. Overall, our toolkit provides an efficient tool

for localizing beaked whales and other vocalizing animals and has the potential to significantly advance our understanding of their behavior and ecology.

## Methods

### Time difference of arrival localization

TDOA localization is a technique that estimates the location of a single sound source by using the arrival times at which the sound is detected on multiple time-synchronized receivers. Typically the source origin time is unknown, but the difference in received times between receiver pairs can be used to determine possible source locations.

There are two forms of TDOA localization that are relevant to our process and are based on array sensor spacing: large-aperture and small-aperture. Large-aperture TDOA localization is used when the distance between the source and receivers is on the same order of magnitude as the distance between the receivers. On the other hand, small-aperture TDOA localization uses receivers that are much closer together than the distance to the source. In this case, the propagation of the signal through the arrays can be approximated as a plane wave.

**Large aperture TDOA.** The TDOA of a signal between two receivers is determined by the distances between the source and each receiver, as shown in Eq (1).

$$\text{TDOA}_{i,j} = \underbrace{\frac{\sqrt{(x_s - x_i)^2 + (y_s - y_i)^2 + (z_s - z_i)^2}}{c}}_{\text{travel time to instrument } i} \cdots$$

$$-\underbrace{\frac{\sqrt{(x_s - x_j)^2 + (y_s - y_j)^2 + (z_s - z_j)^2}}{c}}_{\text{travel time to instrument } j},$$

$$(1)$$

where $x_s$, $y_s$, and $z_s$ are the Cartesian coordinates of the source location, $x_i$, $y_i$, $z_i$, $x_j$, $y_j$, and $z_j$ are the locations of the $i^{\text{th}}$ and $j^{\text{th}}$ receivers, and $c$ is the speed of sound between the source and receivers.

The TDOA from a single pair of receivers produces a hyperboloid of potential source locations, as shown in Fig 1A. The hyperboloid has rotational symmetry about the axis formed by the two receivers. When a detection is received on multiple receiver pairs, the source location can be estimated by finding the intersection of the hyperboloids. However, this approach works best if the receiver pairs are not collinear or somewhat orthogonal to each other and the source is interior to the region defined by the receivers.

**Small aperture TDOA.** When the distance between receivers is much smaller than the distance to the source, the calculation of the TDOA can be simplified as a plane wave propagating through the receiver array. The TDOA is the distance a plane wave travels between the receivers ($d$) divided by the speed of sound ($c$), which can be calculated as the dot product of the vector formed by the hydrophone pair ($\vec{h}_{i,j} = h_j - h_i$) and the unit vector pointing from the source to the receiver ($\vec{s}$). Fig 1B and Eq 2 below demonstrate this calculation.

$$\text{TDOA}_{i,j} = \frac{d_{i,j}}{c} = \frac{\vec{s} \cdot \vec{h}_{i,j}}{c}, \tag{2}$$

For a pair of receivers, this gives a single angle of arrival estimate, resulting in a cone of potential source locations. The hyperboloid shown in Fig 1(A) converges to the cone formed under the plane-wave approximation and introduces negligible error. When multiple small-aperture receiver pairs are combined, the resulting cones intersect along a single line referred

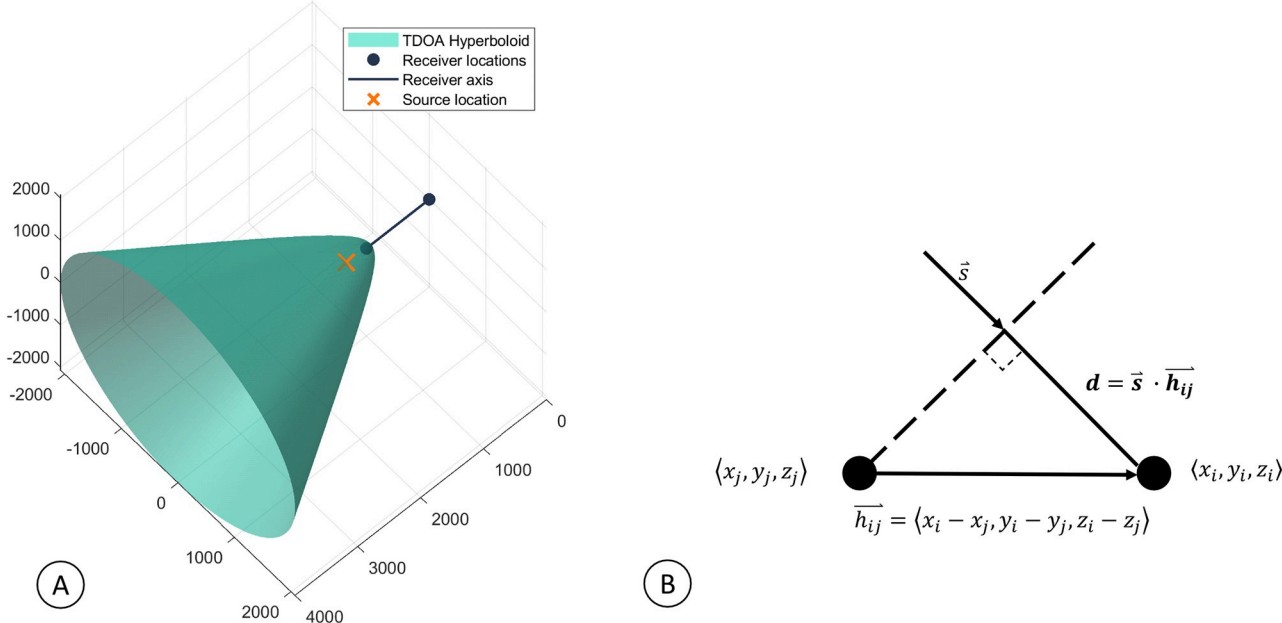

**Fig 1. Time difference of arrival.** Graphical representation of the TDOA for both large and small aperture separation between two sensors. A) Example of a hyperboloid of possible source locations when the TDOA between two widely spaced receivers is known. B) Small-aperture TDOA when the signal's propagation through the array is approximated as a plane wave. The dashed line represents the wave-front, and $\vec{s}$ is the unit vector normal to the wavefront.

to as the Direction of Arrival (DOA). The DOA can be estimated from the TDOAs by placing all hydrophone pairs $\vec{h}_{i,j}$ and their corresponding TDOAs into a system of linear equations, and solving for the unknown values of $\vec{s} = \langle s_x, s_y, s_z \rangle$.

Since the DOA is a unit vector, it can be more intuitively represented by two angles: azimuth and elevation. We define the azimuth (az) as the top-down counter-clockwise horizontal angle, where East is 0°, and North is 90°. The elevation (el) angle is the vertical angle, where 0° is directly down, 90° is horizontal, and 180° is upward toward the sea surface. We convert from $\vec{s}$ to az and el with Eqs 3 and 4:

$$az = \arctan2(-s_y, -s_x),\qquad(3)$$

$$el = 180° - \arccos(-s_z),\qquad(4)$$

where arctan2 is the 2-argument arctangent (`atan2d` in MATLAB). We display these values as pointing from the receiver to the source, which accounts for the negative signs on $s_x$, $s_y$, and $s_z$.

### *Where's Whaledo* software package

The *Where's Whaledo* MATLAB-based software package was designed to help analysts obtain as many animal tracks as possible by providing easy-to-use tools that allow detections to be annotated and tracks of detections to be reconstructed from localized acoustic recordings. This is done using a combination of automated processing and manual annotation of graphical data.

*Where's Whaledo* is specifically designed to accommodate deployments with two volumetric small-aperture arrays and a variable number of single-channel receivers. To perform TDOA localization, the package provides methods to detect signals of interest, determine the time differences of a signal on various receivers, and estimate the most likely source location associated with those time differences. The *Where's Whaledo* toolkit was built in a modular fashion, so each individual step can be adapted to obtain higher precision results or for different instrument configurations. The typical workflow is shown in Fig 2.

**Detection.** Detection steps can be tailored to different species different species using their acoustic parameters. For detecting *Zc* echolocation clicks, we used a fourth-order, zero-phase, high-pass elliptical filter with a cutoff frequency of 20 kHz, a peak-to-peak stop-band ripple of 0.1 dB and a minimum stop-band attenuation of 40 dB. After filtering, waveform sound pressure levels greater than approximately 68 dB re 1 $\mu$Pa$^2$ were identified. Peaks within a ±5 ms window around a larger peak were removed to avoid multiple cycles within a single echolocation click from being counted as separate detections. The remaining peak times were retained as potential click detections.

For the 4-channel data, we cross-correlated the acoustic waveform around each detection across the other receivers in the array to determine the small-aperture TDOA. The TDOA was then converted to an azimuth and elevation using Eqs 2, 3, and 4.

**Association with `brushDOA` tool.** A major challenge in localizing multiple sources with widely-spaced instruments involves identifying the source from which a detection originated. To help analysts with this task, *Where's Whaledo* used an iterative process that combines automated association with analyst manual editing using graphical representations.

A graphical user interface (GUI) tool called `brushDOA` was designed specifically for the purpose of removing false detections, identifying the number of unique sources, and associating detections across the two small-aperture arrays. Using this interface, an analyst can select data points to remove them from the dataset or to assign labels. Collections of detections originating from a single source can be identified by observing the gradual changes in their azimuth and elevation. When azimuth angles from two sources are too similar to differentiate, their elevation angles often provide sufficient separation, and vice-versa. Analysts typically focus their efforts on labeling the array with the most detections and least ambiguity. For example, in Fig 3, array 2 had more detections and clear separation between the various tracks, making it easier for the analyst to identify unique sources and assign labels.

**Association with *Click-Train Correlation* tool.** After labeling one of the arrays, *Click-Train Correlation (CTC)* tool is used to associate detections across the two arrays. The CTC method identifies associations between detections from different instruments by searching for matching patterns [12]. This involves aligning a set of detected clicks in a window of time on different instruments to determine which ones originated from the same source (Fig 4). To accomplish this the method generates click-train vectors $k_i$ by setting a value of one at each detection time:

$$k_i(t) = \begin{cases} 1 \text{ if there is a detection at time } t \text{ on instrument } i, \\ 0 \text{ otherwise.} \end{cases} \quad (5)$$

For recordings from instruments with labeled detections, different vectors of $k_{w,i}[n]$ are generated to include only the echolocation clicks associated with each unique label $w$. For unlabeled data, all detections within the window are used to create the click-train vector. Once the click-train vectors $x_{c,i}$ are generated for each instrument and each whale, they are convolved with a 20 ms wide Hanning window to give some width to the detections. This accounts

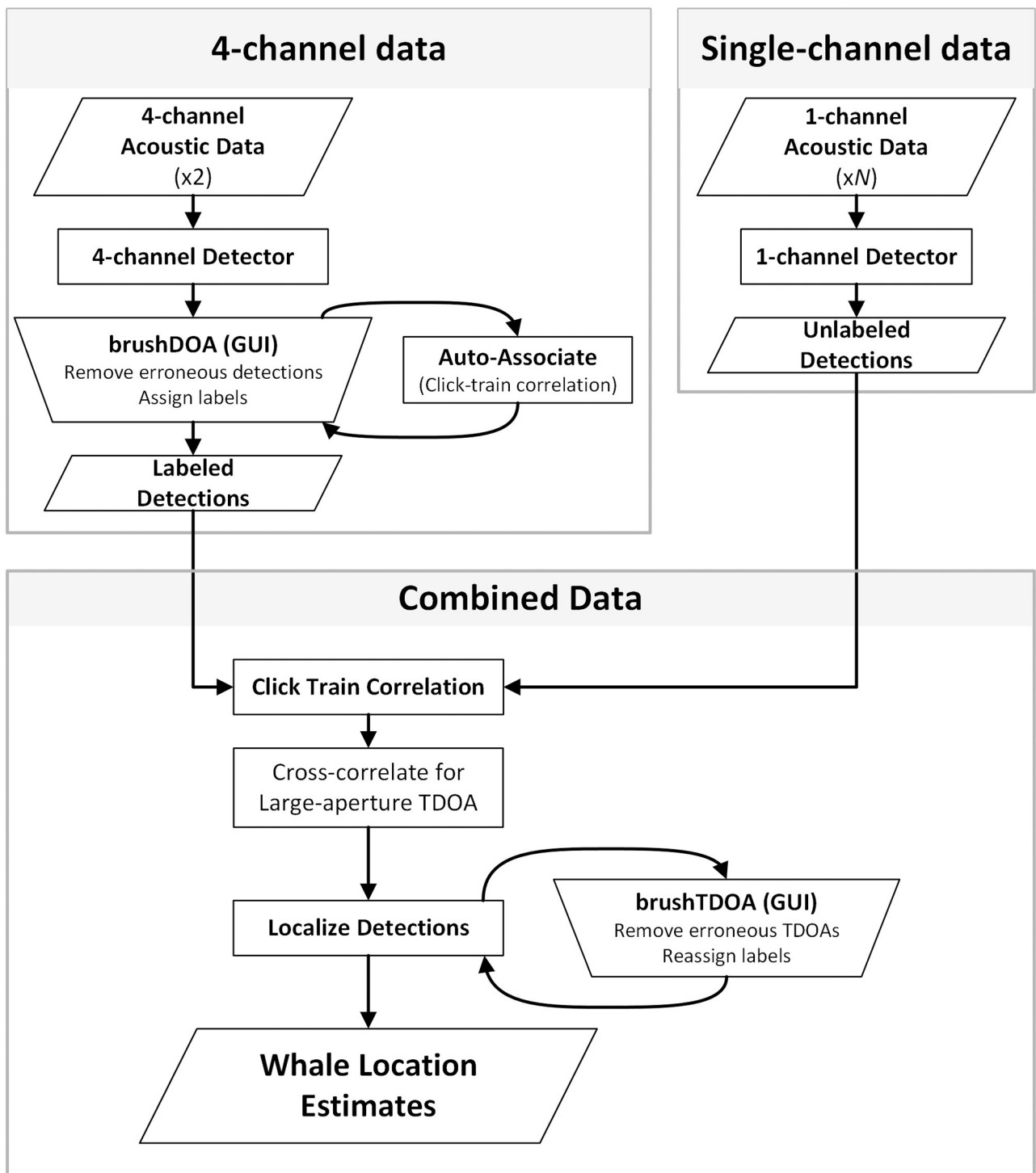

**Fig 2. The *Where's Whaledo* workflow.** The typical workflow used to estimate whale tracks via TDOA localization. The parallelograms indicate data inputs or outputs; the rectangles represent an automated process; the trapezoids indicate a graphical user interface (GUI).

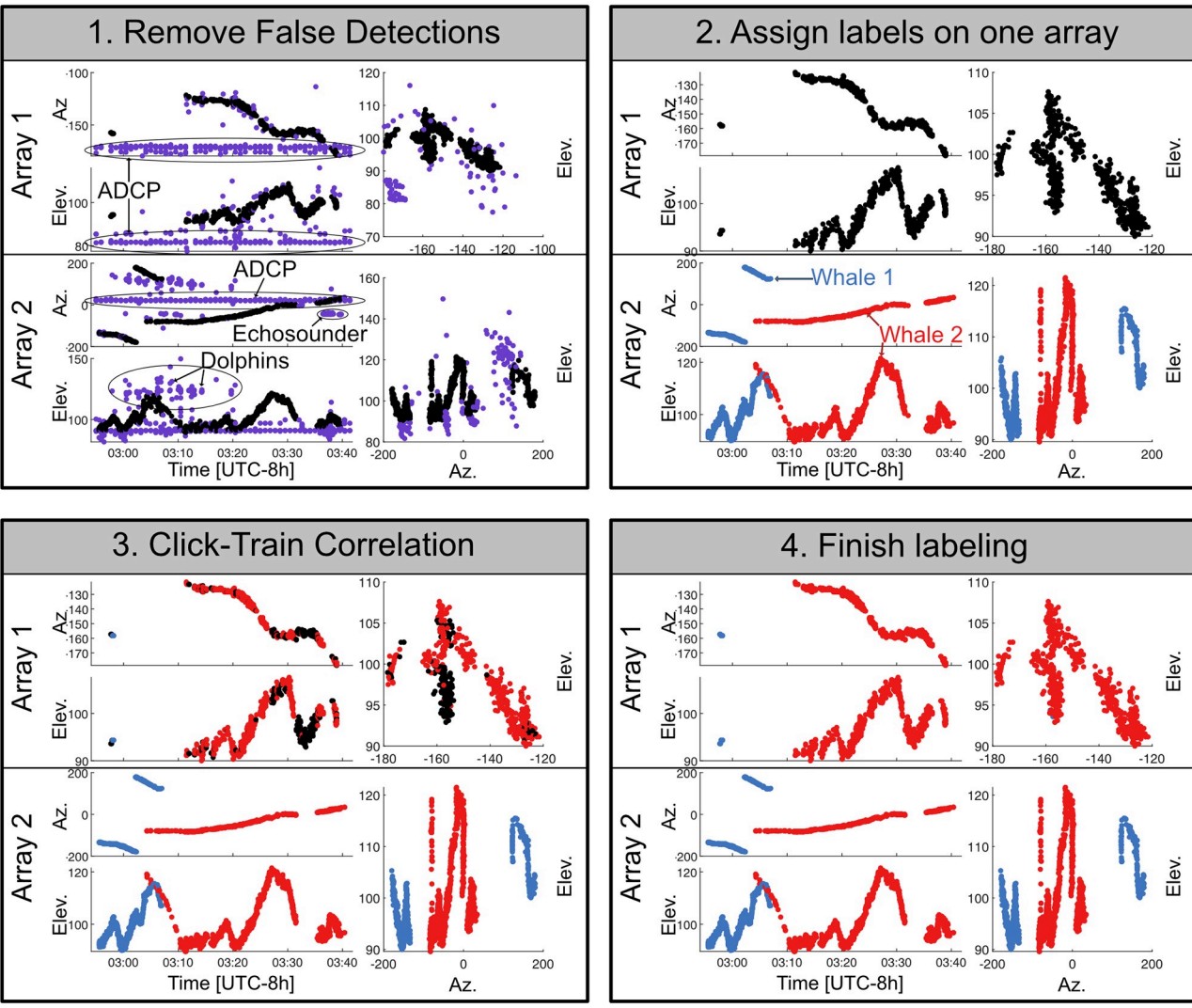

**Fig 3. The `brushDOA` user interface for editing detections on two 4-channel arrays.** The `brushDOA` user interface allows analysts to select detections, remove false detections, and assign color labels to the detections originating from the same source. The interface includes six plots: azimuth vs. time and elevation vs. time for both arrays, and the azimuth vs. elevation for both arrays. Each frame above shows the `brushDOA` interface during four stages of labeling encounters: 1. The analyst removes false detections caused by other nearby sound sources (e.g. ADCP pings, dolphins, instrument noise); 2. The analyst assigns labels on one array to each of the animals present in the encounter using a combination of spatial and temporal separation of detections; 3. Click-Train Correlation is used to automatically associate detections on the labeled array with their corresponding detections on the unlabeled array; and 4. The remaining detections are assigned labels.

for uncertainty in the times of arrival and potential changes in the interval between clicks due to a non-stationary source. The resulting click trains are then cross-correlated, and the location of the peak of the cross-correlation between two click trains gives an estimate of the TDOA ($\tau_w$).

To determine which detections in the unlabeled array are associated with those in the labeled array, the unlabeled detections that align with the labeled detections after being delayed by $\tau_w$ are assumed to originate from the same source and are assigned labels accordingly. However, in cases where both instruments have inadequate detections from the same source, the resulting click-trains may not correlate strongly and may only produce small peaks with no

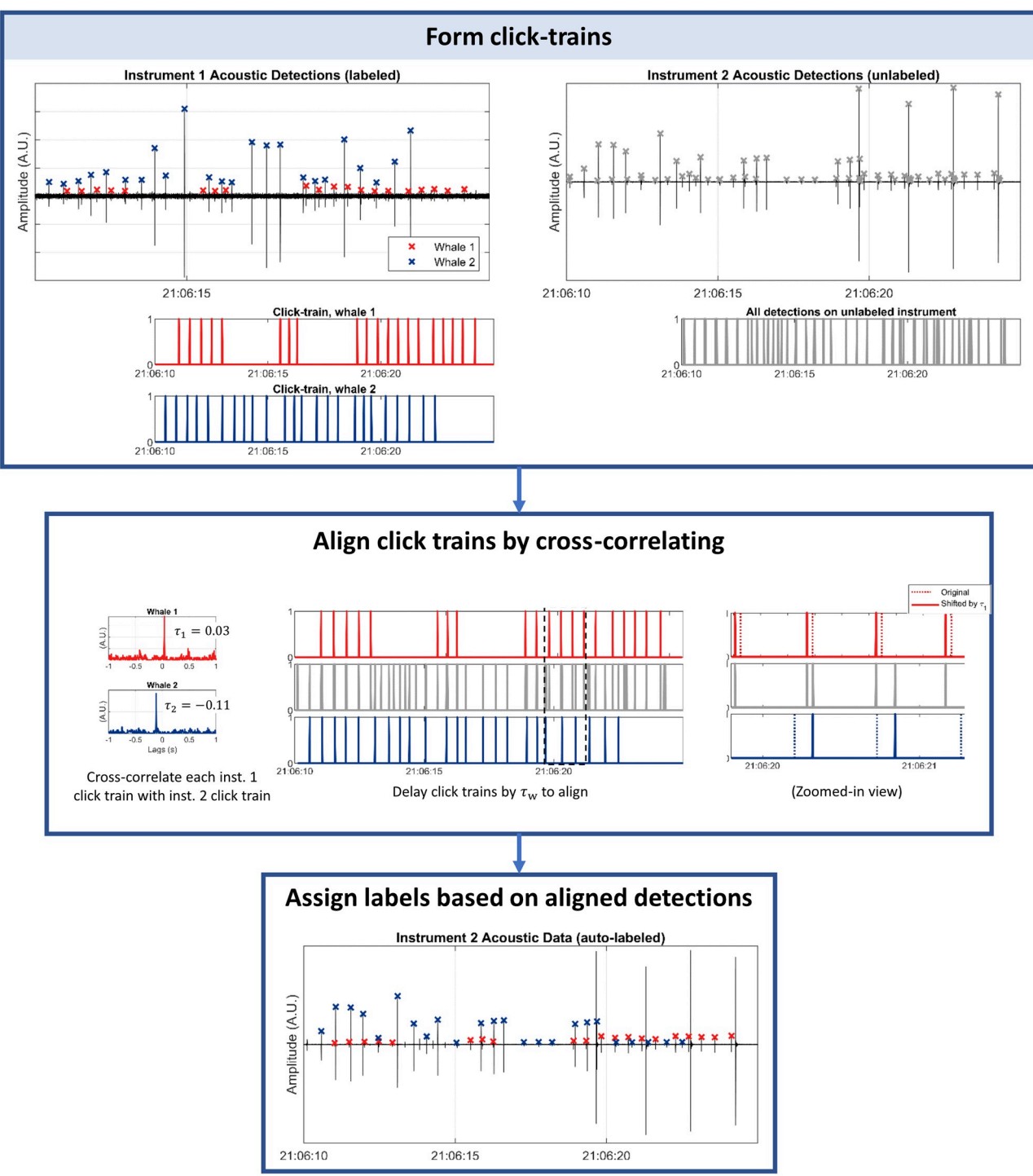

**Fig 4. Click-train correlation.** An example of click-train correlation (CTC) using a window of detections arriving from two sources. The labeled detections (left column) are separated into two click trains, and each is cross-correlated with the unlabeled click train. CTC is used to associate detections across instruments and determine the delay which would align the clicks.

 *Where's Whaledo*: A software toolkit for array localization of animal vocalizations

clear dominant peak that can be used to estimate $\tau_w$. To address this, a condition is set to determine if the click-train correlation has failed due to insufficient detections arriving from the same source. Specifically, if the highest peak in the cross-correlation is not sufficiently higher than all other peaks, the click-train correlation is considered to have failed, and no detections in the window will be assigned labels. As a default parameter, if the second-highest peak is greater than 80% of the value of the highest peak, it is classified as too ambiguous. This percentage is adjustable by the user.

Once a sufficient number of detections are associated with CTC, the analyst uses them to determine which other detections are likely to have originated from the same source based on their azimuths and elevations. However, in some cases, there may be ambiguity in sources as the azimuths and elevations of two sources intersect. These sources can still be associated using CTC from the labeled detections on the other array. Once the labeling process is complete, the analyst can move on to the next phase of localization by incorporating the single-channel detections, as shown in the "Combined Data" box in Fig 2.

The CTC function in *Where's Whaledo* allows adjustment of several parameters including:

- the length of the window used in the click-train correlation,

- the width of the Hanning window convolved with each click train,

- the minimum ratio of the highest peak to the second highest peak in the click-train correlation required to assume the clicks are associated with the same source.

All of these parameters can be adjusted according to the instrument locations, the species of interest, and other features of a deployment. After performing click-train correlation in a window around one detection, the algorithm steps forward to the next detection and repeats the process.

Once the CTC method is used to associate animals across instruments and estimate an approximate TDOA, a fine-scale TDOA measurement is calculated by cross-correlating the acoustic data. To accomplish this, the expected detection times are used to extract the acoustic data around each detection. If there is a mismatched sampling rate, the data are resampled, then filtered and cross-correlated. The time corresponding to the peak in the cross-correlation is used as the precise large-aperture TDOA measurement.

To ensure accuracy, analysts can use a final interactive view to facilitate the removal of erroneous TDOAs or reassign labels to detections that are misassociated in previous steps. This interface is similar to `brushDOA` and typically requires very few changes.

**Monte Carlo bootstrap localization.** To improve localization accuracy, calculate confidence intervals, and combine multiple instrument pairs for each detection, a Monte Carlo Bootstrapping approach is implemented for each detection. First, small gaps in TDOAs are filled in by interpolating between recent detections. Interpolation is only performed when detections are no more than five minutes apart.

Locations are estimated using either one 4-channel array and one single-channel or two 4-channel arrays. For the first case, the intersection between the DOA of the 4-channel and the hyperboloid formed by the large-aperture TDOA between the two instruments is found by calculating the expected large-aperture TDOA at each range step along the DOA line, then taking the range where the error between the expected and measured TDOA is minimized. When localizing with two DOAs, the source location is estimated as the point along one DOA where the distance to any point along the second DOA line is minimized.

A Monte Carlo perturbation method is used to approximate the distribution of locations that can be estimated from each set of TDOAs. Random perturbations are added to the TDOAs using a normally distributed pseudo-random number generator (`randn` in

MATLAB) with variances of $\sigma_{\text{sml}}^2$ (Eq 6) and $\sigma_{\text{lrg}}^2$ (Eq 7) for the small- and large-aperture TDOAs respectively. The process of deriving the variances is presented in S1 Text. DOAs are estimated using the perturbed small-aperture TDOAs, and source locations are estimated for each combination of DOA and large-aperture TDOA available and using both DOAs. This process is repeated 50 times using different random perturbations.

$$\sigma_{\text{sml}} = \sqrt{\left(\frac{\sigma_{H_{k,l}}}{c}\right)^2 + \left(\frac{\|H_{k,l}\|}{c}\right)^2\left(\frac{1}{100^2}\sigma_{h_i}^2 + \sigma_{\text{ray}}^2\right) + \left(\frac{\text{TDOA}(k,l)_{\text{calc}}}{c}\right)^2\sigma_c^2 + \sigma_{\text{xcorr}}^2}. \quad (6)$$

$$\sigma_{\text{lrg}} = \sqrt{\frac{\sigma_{h_i}^2 + \sigma_{h_j}^2}{c^2} + \left(\text{TDOA}(i,j)_{\text{calc}}\right)^2\frac{\sigma_c^2}{c^2} + \sigma_{\text{travel\ time}}^2 + \sigma_{\text{drift}}^2 + \sigma_{\text{xcorr}}^2}. \quad (7)$$

Each Monte Carlo location estimate is stored to produce a distribution of potential source locations for one detection. Location estimates are assigned a weight equal to the inverse of the variance of the location estimates using the same combination of instruments. A bootstrapping estimate of the weighted average is used to produce the final source location estimate [43, 44]. This involves randomly replacing location estimates with other estimates in the distribution and recalculating the weighted average source location estimate (resampling with replacement). Resampling is repeated 50 times, and the average of the resampled weighted average estimates is used as the final source location estimate. The 95% confidence intervals are estimated using a Studentized bootstrap method [43, 45].

**Alternative localization approach—DOA intersect.** For deployments localizing with two volumetric arrays and no single-channel instruments, the localization process can be linearized and performed much faster. The process is identical to the 4-channel data box in Fig 2, but rather than incorporating the single-channels with click train correlation, the labeled detections from each 4-channel were localized by finding the closest point of intersection between the two DOA lines. This is done by solving the system of equations relating the source location to the directions of arrival,

$$\vec{s}_1 r_1 + h_1 = \vec{s}_2 r_2 + h_2 = g_{i,j,k}, \quad (8)$$

where $\vec{s}_n$ is the unit vector representing the DOA line for the $n^{\text{th}}$ array, $r_n$ is the range from the $n^{\text{th}}$ array to the source, and $h_n$ are the Cartesian coordinates of the $n^{\text{th}}$ array location (see Fig 5 for a visualization). By finding the values for $r_1$ and $r_2$ which minimize $(\vec{s}_1 r_1 + h_1) - (\vec{s}_2 r_2 + h_2)$, we can estimate the point along each DOA line where the lines are closest to intersecting. To do this, a 2x3 matrix $\mathbf{S} = [\vec{s}_1; -\vec{s}_2]$ is constructed and $\mathbf{R} = [r_1; r_2]$ is solved for using MATLAB's "backslash" (or `mldivide`) function (Eq 9). This results in two estimates: $g_1 = \vec{s}_1 r_1 + h_1$ and $g_2 = \vec{s}_2 r_2 + h_2$. The final source estimate is the average of $g_1$ and $g_2$.

$$\mathbf{R} = \mathbf{S} \setminus (h_2 - h_1). \quad (9)$$

The confidence intervals for this method of localization were obtained using the jackknife variance estimator [46]. One TDOA and its associated receiver pair is removed from the DOA estimation, and a new DOA is estimated using the remaining five TDOAs. A new whale location is estimated using the intersection point of the newly obtained DOA and the DOA of the other array using Eq 9. This is repeated for each receiver pair, removing one TDOA and localizing with the remaining 11, until 12 different whale location estimates have been produced.

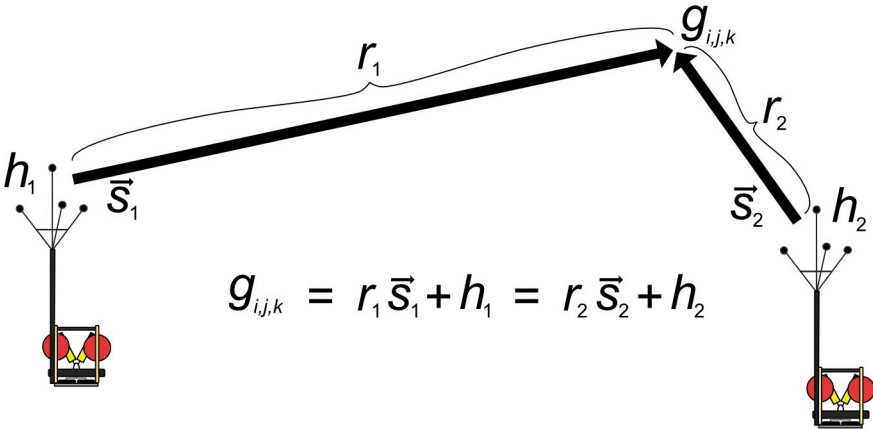

$$g_{i,j,k} = r_1 \vec{s}_1 + h_1 = r_2 \vec{s}_2 + h_2$$

**Fig 5. Visualization of the DOA intersect localization method.** An alternative method of localization when detections are present on both 4-channel arrays is to find the intersection of the two DOA lines, $\vec{s}_1$ and $\vec{s}_2$. This is done by solving the Eq 8 using MATLAB's `mldivide` function (Eq 9).

The variance of these location estimates is determined and used in the inverse Student's T distribution to estimate the 95% confidence intervals.

## Case study—Tanner Basin

Our demonstration of *Where's Whaledo* localizes *Zc* using a dataset collected during a four-month deployment about 200 km southwest of Los Angeles, California, in the Tanner Basin, known for its *Zc* presence (Fig 6). Four High-frequency Acoustic Recording Packages, or HARPs [20, 47] were deployed from March 16th to June 11th, 2018. The north and south HARPs each had a single omnidirectional hydrophone with a sampling rate of 200 kHz moored approximately 10 m above the seafloor. The east and west HARPs each had volumetric arrays of four omnidirectional hydrophones in a tetrahedron configuration with $\approx$ 1 m spacing between hydrophones. The 4-channel arrays had a sampling rate of 100 kHz and sat $\approx$ 6 m above the seafloor on a rigid mast. The distance between each HARP was between 470 and 1075 m. Bathymetry data used for plotting obtained from the Global Multi-Resolution Topography (GMRT) map tool (https://www.gmrt.org/GMRTMapTool) [48].

The data used in the case studies (both unedited detections and final track reconstructions) are available in the Dryad data repository: DOI:10.5061/dryad.c866t1gfj [49].

**Oceanographic conditions and instrument locations.** TDOA localization requires knowledge of receiver locations and the properties of the medium of propagation that affect travel times. The speed of sound in water depends on various oceanographic conditions, such as temperature, pressure, and salinity, resulting in both temporal and spatial variation in sound speed [50–52]. However, to simplify computation, a constant sound speed was used for our case study. This approximation is generally acceptable at close ranges. To quantify the error introduced by the constant sound speed approximation, we estimated the variations in sound speed using a CTD (Conductivity, Temperature, Depth) profiler mounted at the study site. Empirical relationships between sound speed, temperature, salinity, and depth were used to estimate sound speed from the CTD measurements [50–52]. The uncertainty in the assumed sound speed is accounted for in the overall uncertainty of the localization estimates. Further details on the uncertainty calculations can be found in S1 Text.

Each instrument is equipped with an Edgetech acoustic release that can emit an acoustic ping in response to a ping received from a transducer on the ship. The two-way travel time of

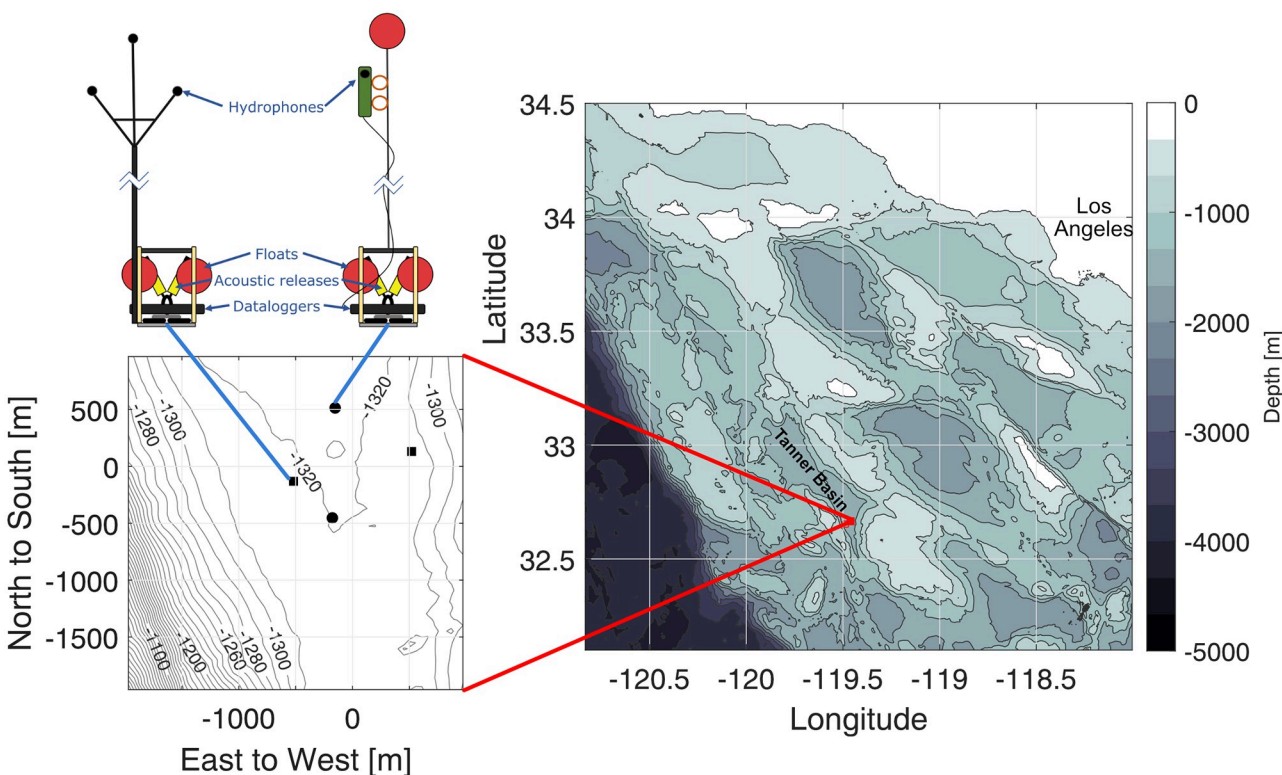

**Fig 6. Study site.** The case study site where *Zc* tracks were reconstructed using the *Where's Whaledo* MATLAB toolkit. Site is in Tanner Basin, ≈ 200 km southwest of Los Angeles, California. Two instrument types were used: single channel instruments (black circles on the left plot) and 4-channels (black squares). Bathymetry data from Global Multi-Resolution Topography (GMRT) [48].

these acoustic signals from various ship locations is used to estimate the positions of the instruments. The uncertainty in instrument position is incorporated into the overall uncertainty and is discussed further in S1 Text.

To determine the relative positions of the hydrophones in the small-aperture arrays, we use the plane-wave approximation as shown in Eq 2. Instead of relying on a narrow-band ping, we used the broadband engine noise emanating from the ship during the instrument localization period. The engine noise is bandpass filtered and cross-correlated to estimate the TDOA in one-second bins. The TDOA's and the ship location for each one-second bin (obtained from the ship's GPS system) are put into a system of equations using Eq 2 to solve for the relative hydrophone positions within the array.

**Clock synchronization.** Ensuring clock synchronization is essential for combining data from various receivers used in localization. While all the receivers within each small aperture array were synchronized, the large aperture array required a two-step process to correct for clock drift. Initially, we synchronized the clocks using the pings transmitted by each instrument's acoustic release during instrument localization. Then, we used the pings from an Acoustic Doppler Current Profiler (ADCP) which was deployed concurrently with our instruments and transmitted a 75 kHz ping approximately every 60 seconds to synchronize the clocks for the remainder of the deployment. Well 75 kHz is above the Nyquist frequency of the 4-channel instruments (50 kHz), the pings were recorded as aliased signals of 25 kHz and could still be used for clock synchronization.

Each instrument's acoustic release was enabled only during the period when it was being localized. Instrument localization was performed over the course of seven hours, and each acoustic release was enabled for between one and two hours. The pings were detected with a narrowband filter and a threshold. Due to the consistency of the amplitude of the pings, a different threshold was used for each instrument which was well above the noise levels at this frequency but had a near-zero probability of missed detection. The TDOA was calculated by cross-correlating the pings detected on each instrument. False detections produced TDOAs that significantly deviated from the true TDOAs and were manually removed. The clock drifts were calculated as the values that minimized the errors between the expected TDOAs (based on instrument locations and sound speed) and the calculated TDOAs.

The ADCP pinged approximately every minute at 75 kHz. Since the 4-channel HARPs had a sampling rate of 100 kHz, the aliased frequency of 25 kHz was used to calculate the TDOA of the ADCP pings. The single-channel data were downsampled from 200 kHz to 100 kHz to deliberately alias the ADCP pings. The TDOA was then calculated by cross-correlating the detected ADCP pings for the entire deployment. The relative clock drifts between each instrument pair was then estimated as the change from the expected TDOA (based on the TDOA of the ADCP pings calculated during localization). A fifth-order polynomial fit was applied to the resulting clock drift estimates to simplify correcting for clock drift during localization.

## Dryad DOI

[10.5061/dryad.c866t1gfj](10.5061/dryad.c866t1gfj)

## Results

In our case study dataset, we used a specialized beaked whale detector in tandem with DetEdit [53] to identify 600 separate time periods containing *Zc* detections. Of these initial periods with detections, 107 contained detections with a high enough SNR and were in close enough proximity to the instruments for analysts to identify unique individuals in the encounter using `brushDOA`. However, many of these individual tracks had too few detections to be reliably localized; encounters that lasted less than 5 minutes or contained fewer than 300 detected clicks were removed from analysis, and ultimately approximately 90 encounters contained a sufficient number of localized detections in succession to be considered usable tracks. These encounters contained between one and six uniquely identifiable individual animals.

We demonstrate our approach with three examples. The first example is the simplest case, where source association is unambiguous, and tracks can be obtained quickly and easily. The second example is an encounter with six whales where source association was more challenging due to the number of whales vocalizing simultaneously and their proximity to each other. In the last example, a large pod of vocalizing dolphins obscured the beaked whale vocalizations, but we were still able to obtain tracks of two individual *Zc* using DOA information and click-train correlation.

### Example 1—Simple source association

In this example, two whales were observed that exhibited both spatial and temporal separation, facilitating a straightforward association of clicks to each source. The encounter occurred on June 11, 2018, as the whales both approached the acoustic array. The first whale swam to the northeast, passing just west of the array, while the second whale swam northward, moving directly into the center of the array (Fig 7). The distinct spatial and temporal gap between the whales allowed for unambiguous source association. Click-train correlation performed well on both tracks, further ensuring accurate source associations. Additionally, sporadic detections of

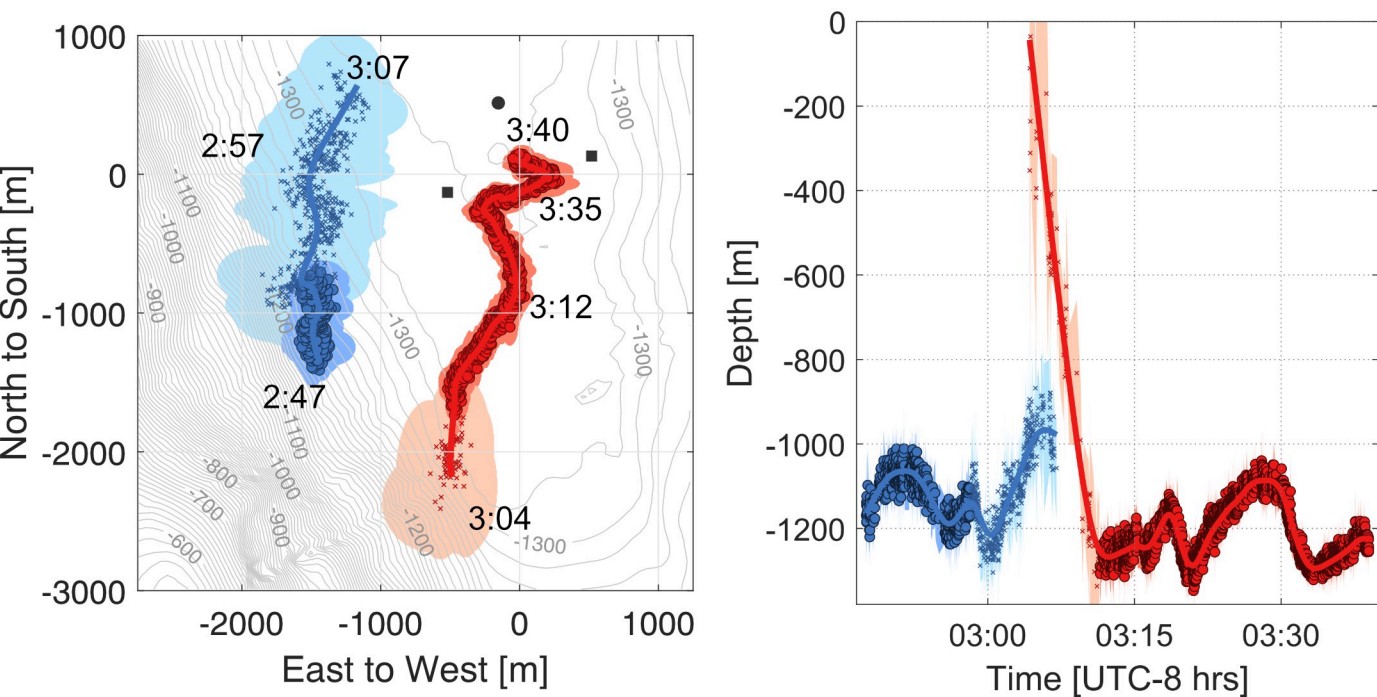

**Fig 7. *Zc* track reconstructions with clear source association.** The left panel is a map view with time annotations along two separate animal tracks, and the right panel shows the animals' depth versus time. The colors represent different whales, and the semi-transparent shading represents their 95% confidence intervals. Points with circles are localized with two 4-channel instruments, whereas points with "x" were detected on only one 4-channel and one or two single-channels, Confidence intervals vary due to differences in the number of instruments used to localize, the position of the whale, or the precision and accuracy of the TDOAs.

a possible third and fourth whale occurred during this time period, but they were insufficient to establish reliable track formations.

Both whales in this example exhibited a dive descent at the beginning of their tracks. The first whale was positioned more than 3000 m from the center of the array. Since errors in DOA angle estimates scale with range, this leads to larger confidence intervals when compared to the much closer second whale. During the dive phase of the second whale, detections were present only on the west 4-channel and one or both of the single channels. Once the whale reached foraging depth, all four instruments had a significant number of detections, allowing for optimal track reconstruction.

### Example 2—Large group size

An encounter involving at least five *Zc* was identified on April 29, 2018 (Fig 8). The whales were observed in two distinct clusters: a first group of three whales swimming from the south and east toward the center of the array at the beginning of the encounter (red, blue, and yellow), and a second pair following about 10 minutes behind from the same direction (purple and green).

### Example 3—*Zc* co-occurance with dolphins

An encounter on April 22, 2018 consisted of a group of two *Zc* echolocating simultaneously with a large pod of dolphins (Fig 9). The overlap in the frequencies of both dolphin and *Zc* clicks led to a high number of false detections. It is worth noting that dolphin dive depths are much shallower than *Zc*, resulting in the elevation angles of the dolphin detections being closer

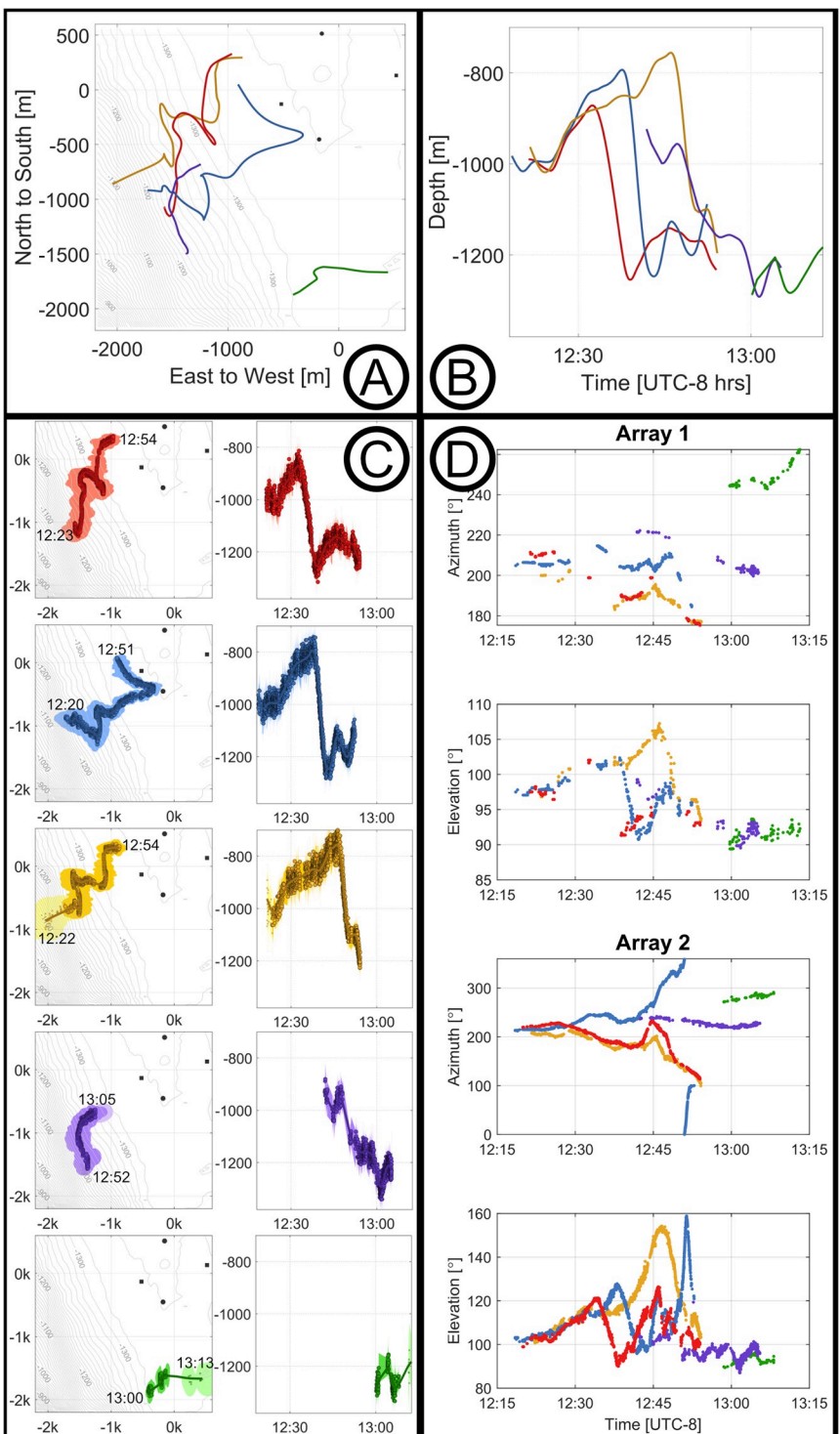

**Fig 8. *Zc* tracks with large group sizes.** An encounter with five whales vocalizing concurrently. Panels A and B show the map view and the depth vs. time of the track estimates of all five animals, where the colors correspond to the same detections shown in the other panels. Panel C shows the map view and depth vs. time views for each individual separately, where the different colors represent different whales and the semi-transparent shading represents their 95% confidence intervals. Panel D shows the labeled azimuths and elevation angles of each of the animals in the encounter.

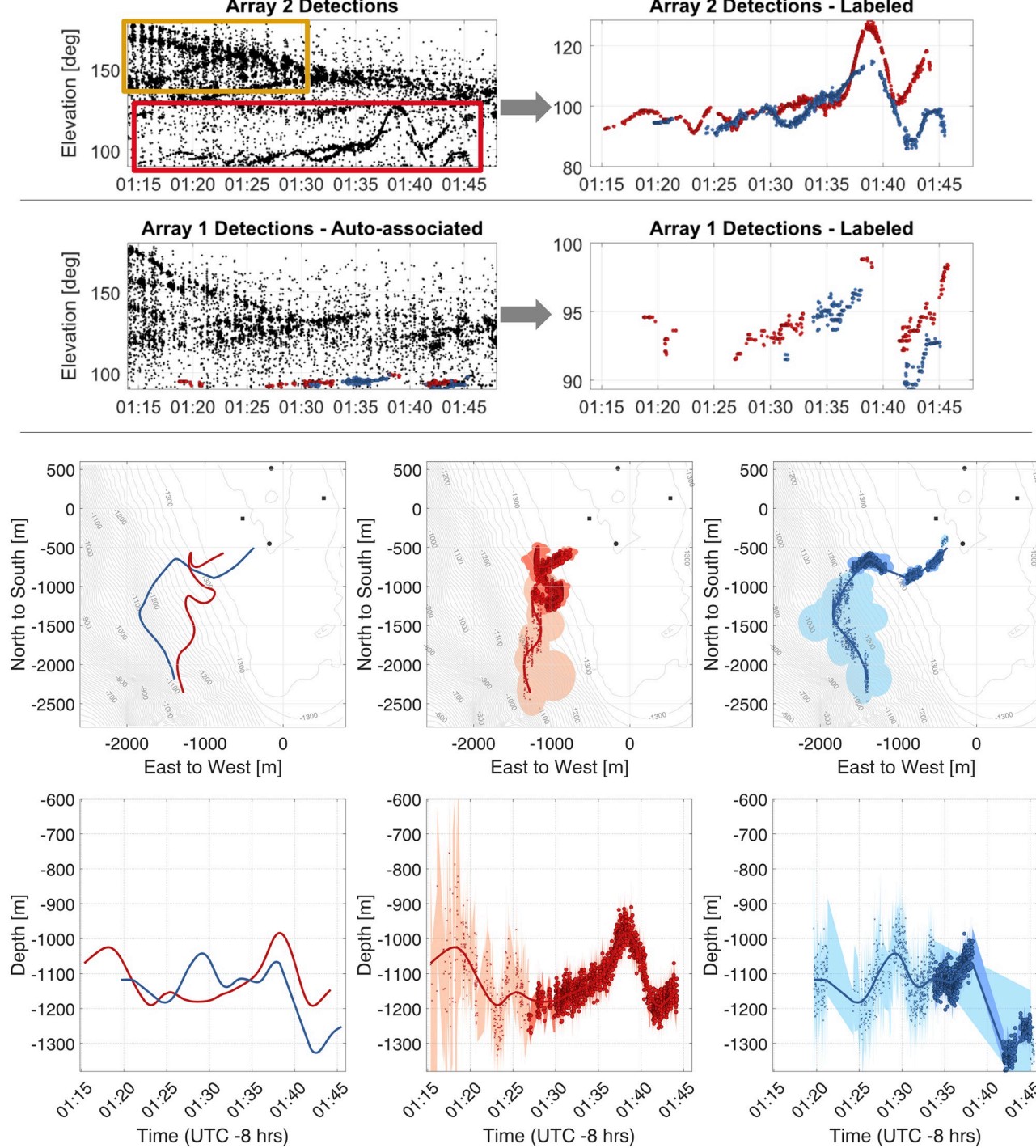

**Fig 9. Reconstructing *Zc* tracks in the presence of false-detections.** Top left panel shows array-two detections, including: (yellow box) echolocating dolphins and (red box) two echolocating *Zc*. Upper right panel illustrates removal of dolphin detections, due to their higher elevation angles, periodicity (where detections fade in and out on an ≈ 1 min cycle), and "fuzziness" (where multiple dolphin clicks present in one window gave erroneous DOAs). Middle panels show array-one detections (left) before and (right) after dolphin echolocation removal. Lower panels show maps with tracks of (left) both *Zc* and (middle and right) individual animals.

to 180˚ than the *Zc* detections. However, when the dolphin group sizes are large, as for this case, multiple individuals' clicks arrive within the allowed small-aperture TDOAs, and their cross-correlations frequently produce erroneous DOA estimates. Consequently, the resulting DOA plots appear cluttered with detections seemingly coming from all directions, including the seafloor.

Nevertheless, the *Zc* clicks produce reliable TDOA estimates, allowing for their visual identification in the DOA plots. Dolphin detections could also be identified by their periodicity, with clicks occurring in clusters that faded in and out every few minutes. This characteristic made them easier to identify and remove from the analysis.

In this instance, identifiable *Zc* tracks were present in array two, while they were less distinct in array one. Array two was therefore cleaned and labeled, followed by click-train correlation to determine the most likely *Zc* clicks on array one (Fig 9). The resulting *Zc* tracks approach the array center from the southwest, apparently in a coordinated manner.

## Discussion

This study demonstrates the utility of *Where's Whaledo* as a tool for reconstructing tracks using passive acoustic localization. We were able to obtain 90 reliable tracks from a four-month deployment offshore of Southern California. The process has the potential to be applied to similar deployments, and further development of the software could expand its usefulness to other receiver configurations, environments, and species of interest.

Identifying potential tracks and removing erroneous or unreliable detections can be done with the `brushDOA` GUI, which allows analysts to efficiently identify and annotate detections arriving from the same source on a small aperture array. Automated source association between widely spaced receivers is performed with click-train correlation, which searches for patterns of clicks arriving from one source in the various receivers. Once detections are correctly associated, they can be cross-correlated to determine the fine-scale TDOA, then localized using maximum likelihood comparison with a TDOA model.

A primary localization challenge is categorizing clicks by individual animals. When the animals are far apart, individuals can be successfully identified in the Azimuth/elevation plots. This was occasionally challenging with *Zc*, but for most encounters, distinct tracks could be identified on at least one of the small-aperture arrays. Calculating the TDOA on the small-aperture arrays by cross-correlating a window of time around a detection assumes only one detection within the window. For species with more individuals or whose interval between clicks is shorter than the maximum possible TDOA like some dolphin species, this may not hold, and an alternative method for identifying sources would be necessary. Click-train correlation can be effective in finding patterns of clicks on separate instruments, but may not work for other species with less unique click patterns or where detections are too sparse for adequate correlation. In these cases, analysts may rely on identifying periods of simultaneous elevation change on both arrays or incorporate other methods to associate detections with sources.

In this study, simultaneous occurrences of *Zc* and delphinids presented challenges for tracking beaked whales. One solution would be to use a more sophisticated detector that better differentiated between each species' vocalizations, for instance using measurements of peak frequency and number of cycles within a click to separate species, or using a machine-learning based detector [54, 55]. However, due to identifiable patterns in the DOA plots, such as higher elevation angles, periodicity in vocalizations, and a high number of erroneous DOA estimates, dolphin detections were frequently able to be manually removed by analysts while *Zc* detections were retained.

The tracks obtained from our approach often contain spatial offsets in clusters of detections arriving from the same source, causing the path to appear bifurcated. This is generally due to different combinations of instruments detecting the echolocation pulses. The most reliable detections were those that were detected on both 4-channel instruments. Due to the distance between the two 4-channels in this deployment (1070 m) and the highly directional nature of *Zc* echolocation clicks, many detections were only present on one of the 4-channels. Placing the arrays closer together would increase the number of clicks detected on both 4-channels. However, this would decrease the range at which reliable localizations were possible. Therefore, finding the optimal balance between the distance between the arrays and the number of clicks detected on both 4-channel instruments is crucial. Click directionality also greatly influences the spatial distribution of tracks obtained, since an encounter is far more likely to be tracked when the animal is facing multiple instruments. This may introduce bias into the types of track obtained at a given site, and a thorough analysis of this spatial bias should be performed for each deployment.

*Where's Whaledo* was developed for and tested specifically on deployments with two 4-channel HARPs and a varying number of single-channel instruments. With some adaptations, *Where's Whaledo* could prove useful with varying instrument configurations, such as large-aperture only or linear arrays. As of publication, the detector and TDOA estimator included on the *Where's Whaledo* GitHub page were developed for *Zc*, but the software could be expanded to be used for localizing other sources, such as baleen whales or anthropogenic sounds.

To improve *Where's Whaledo*, a more advanced detector could be used to incorporate low SNR clicks without generating false detections. Jang *et al*. [16] implemented a Generalized Cross-correlation detector on the same dataset, which was effective in removing most false detections caused by repeated instrument sounds. Additionally, Jang et al [16] used a multi-target tracking (MTT) algorithm to reconstruct *Zc* tracks using the two small aperture volumetric arrays. Components of this algorithm could be incorporated into *Where's Whaledo* to automate the removal of false TDOA measurements and improve source association. By incorporating estimates of an animal's swim speed into localizations, the reliability of track reconstructions could be further enhanced [14, 16].

## Conclusion

Passive acoustic localization is a powerful way to track animal movement, which can provide valuable insights into animal behavior and the parameters needed for density and distribution measurements. Several previous studies have demonstrated the capability of using TDOA localization of cetacean vocalizations to reconstruct their tracks. Some common challenges may limit the number of tracks obtained, including efficiently identifying potential tracks in large datasets, identifying the number of sources, and associating detections to the appropriate source. The *Where's Whaledo* toolkit provides an efficient and reliable workflow for TDOA localization of odontocete echolocation clicks. The toolkit is designed for deployments of hydrophones containing a combination of small-aperture volumetric arrays and single-channel instruments. *Where's Whaledo* includes a number of functions and GUIs to aid in the process of identifying separate sources, associating detections to each source, removing erroneous or unreliable detections, and estimating the most likely whale position from the TDOAs.

We demonstrate the utility of *Where's Whaledo* by localizing *Zc* echolocation clicks in the Tanner Basin. In the four-month dataset, tracks were reconstructed for ≈ 90 individual whales, with group sizes ranging from one to six individuals. Track reconstructions were successfully performed in the presence of significant masking due to dolphin echolocation clicks

and in situations where animals were in close proximity. With some adaptations, *Where's Whaledo* could be configured to work with a variety of receiver configurations, environments, and species of interest.

## Supporting information

**S1 Text. Derivation of TDOA uncertainties $\sigma_{sml}$ and $\sigma_{lrg}$.**
(PDF)

## Acknowledgments

Special thanks to: Bruce Cornuelle for providing guidance on efficient computations and other expertise; Junsu Jang for testing Kalman and particle filter implementations and general feedback; Bayleigh Coleman, Alma Leon, and Ryan Parkes for beta-testing earlier versions of *Where's Whaledo* software. Ana Mae Shickich, Grace Randall, and Lauren Baggett for testing the current version and providing valuable feedback. *Where's Whaledo* naming credit goes to Margaret Morris. Additional thanks Robert Headrick and Chip Johnson for their support in obtaining funding.

## Author Contributions

**Conceptualization:** Eric R. Snyder, Simone Baumann-Pickering, Sean M. Wiggins, John A. Hildebrand.

**Data curation:** Eric R. Snyder, Alba Solsona-Berga, Simone Baumann-Pickering, Sean M. Wiggins, John A. Hildebrand.

**Formal analysis:** Eric R. Snyder, Sean M. Wiggins, John A. Hildebrand.

**Funding acquisition:** Simone Baumann-Pickering, John A. Hildebrand.

**Investigation:** Eric R. Snyder, Alba Solsona-Berga, Kait E. Frasier, Sean M. Wiggins, John A. Hildebrand.

**Methodology:** Eric R. Snyder, Alba Solsona-Berga, Kait E. Frasier, Sean M. Wiggins, John A. Hildebrand.

**Project administration:** Simone Baumann-Pickering, Sean M. Wiggins, John A. Hildebrand.

**Resources:** Sean M. Wiggins, John A. Hildebrand.

**Software:** Eric R. Snyder, Alba Solsona-Berga, Kait E. Frasier, Sean M. Wiggins.

**Supervision:** Simone Baumann-Pickering, Kait E. Frasier, Sean M. Wiggins, John A. Hildebrand.

**Validation:** Eric R. Snyder, Simone Baumann-Pickering, Sean M. Wiggins, John A. Hildebrand.

**Visualization:** Eric R. Snyder, Simone Baumann-Pickering, John A. Hildebrand.

**Writing – original draft:** Eric R. Snyder.

**Writing – review & editing:** Eric R. Snyder, Alba Solsona-Berga, Simone Baumann-Pickering, Kait E. Frasier, Sean M. Wiggins, John A. Hildebrand.

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
