## [Decision Letter · Decision Letter 0]

22 Jan 2024

Dear Mr. Snyder,

Thank you very much for submitting your manuscript "Where's Whaledo: A software toolkit for array localization of animal vocalizations." for consideration at PLOS Computational Biology. As with all papers reviewed by the journal, your manuscript was reviewed by members of the editorial board and by several independent reviewers. The reviewers appreciated the attention to an important topic. Based on the reviews, we are likely to accept this manuscript for publication, providing that you modify the manuscript according to the review recommendations.

Dear Eric Snyder,

I am sorry for the long delay in reviewing your article but it has been particularly difficult to find willing reviewers. I have to say that this type of paper which involves a mixture of methods, a "guide" for a software package and various illustrations is not something that I have seen published in PLOS Comp Biology where most papers are less applied. This said it is clear that this paper is well done and will be useful to your community. It might therefore be a good pilot for these more method oriented papers in Plos Comp Biology.

I made a few minor comments of my own:

L 34 . extra parenthese.

L. 55. Did you mean “both small-aperture DOA estimates and large-aperture TDOAs”? they are both TDOA estimates at first no?

Equation 1. Don’t you need a - in the equation to calculate the difference in arrival times?

Sincerely,

Frédéric E. Theunissen

Academic Editor

PLOS Computational Biology

Zhaolei Zhang

Section Editor

PLOS Computational Biology

Dear Eric Snyder,

I am sorry for the long delay in reviewing your article but it has been particularly difficult to find willing reviewers. I have to say that this type of paper which involves a mixture of methods, a "guide" for a software package and various illustrations is not something that I have seen published in PLOS Comp Biology where most papers are less applied. This said it is clear that this paper is well done and will be useful to your community. It might therefore be a good pilot for these more method oriented papers in Plos Comp Biology.

I made a few minor comments of my own:

L 34 . extra parenthese.

L. 55. Did you mean “both small-aperture DOA estimates and large-aperture TDOAs”? they are both TDOA estimates at first no?

Equation 1. Don’t you need a - in the equation to calculate the difference in arrival times?

Reviewer's Responses to Questions

**Comments to the Authors:**

Reviewer #1: I enjoyed reading this manuscript. Overall it is of good quality and am happy to recommend it for publication after some minor revisions. In addition to the comments from the attached pdf, here are a few items that may require attention:

1. Has this tool been tested on data from instruments other than HARPs? If not, what considerations should be made? While this seems to be a great tool for 4-channel HARP data, showing it is easily adaptable to other datasets from different tetrahedral arrays would make it more attractive.

2. The are few mentions of the directionality issue in the manuscript. First, it should be mentioned that the receivers are omnidirectional (I believe they are?). Second, based on the directionality of beaked whale clicks, I assume you can mostly track whales swimming towards the array vs. whales swimming away from the array. If so, this should be mentioned at least in the discussion.

3. Many localization techniques for odontocetes use multipath. While it is my understanding you don't use these at all, this approach should be mentioned in the literature review, and your discussion could also talk about how using multipath could help improve your current approach. Especially to fill gaps where an animal is not detected on enough instruments to localize using TDOA from various sensors.

4. It would be nice in Figure 7 / Example 2 to show the TDOA tracks and how you are able to separate the six tracks and associate them to the same source across the different sensors. This example seems to be the most challenging and it would be nice to see in more detail how it is done.

5. In the abstract and conclusion you mention how tracking can be useful for things like population density, behavior etc. It would be nice to have a few sentences in the discussion explaining how. Or even better use your results to point out specific things that would be useful for population density, behavior etc.

**Have the authors made all data and (if applicable) computational code underlying the findings in their manuscript fully available?**

Reviewer #1: Yes

PLOS authors have the option to publish the peer review history of their article (what does this mean?). If published, this will include your full peer review and any attached files.

Reviewer #1: **Yes: **Ludovic Tenorio-Hallé

Figure Files:

Data Requirements:

Reproducibility:

References:

---

## [Editor Report · Decision Letter 1]

30 Apr 2024

Dear Mr. Snyder,

We are pleased to inform you that your manuscript 'Where's Whaledo: A software toolkit for array localization of animal vocalizations.' has been provisionally accepted for publication in PLOS Computational Biology.

Best regards,

Frédéric E. Theunissen

Academic Editor

PLOS Computational Biology

Zhaolei Zhang

Section Editor

PLOS Computational Biology

---

## [Editor Report · Acceptance letter]

10 May 2024

PCOMPBIOL-D-23-01359R1 

Where's Whaledo: A software toolkit for array localization of animal vocalizations.

Dear Dr Snyder,

I am pleased to inform you that your manuscript has been formally accepted for publication in PLOS Computational Biology. Your manuscript is now with our production department and you will be notified of the publication date in due course.

With kind regards,

Zsofia Freund
